# Self-Healing Iron Oxide Polyelectrolyte Nanocomposites: Influence of Particle Agglomeration and Water on Mechanical Properties

**DOI:** 10.3390/nano13232983

**Published:** 2023-11-21

**Authors:** Bastian Oberhausen, Ajda Plohl, Bart-Jan Niebuur, Stefan Diebels, Anne Jung, Tobias Kraus, Guido Kickelbick

**Affiliations:** 1Inorganic Solid-State Chemistry, Saarland University, Campus, Building C4 1, 66123 Saarbrücken, Germany; bastian.oberhausen@uni-saarland.de (B.O.); ajda.plohl@uni-saarland.de (A.P.); 2INM—Leibniz-Institute for New Materials, Campus, D2 2, 66123 Saarbrücken, Germanytobias.kraus@leibniz-inm.de (T.K.); 3Applied Mechanics, Saarland University, Campus, Building A4 2, 66123 Saarbrücken, Germany; s.diebels@mx.uni-saarland.de; 4Protective Systems, Helmut-Schmidt-University/University of the Federal Armed Forces Hamburg, Holstenhofweg 85, 22043 Hamburg, Germany; anne.jung@hsu-hh.de; 5Colloid and Interface Chemistry, Saarland University, Campus, Building D2 2, 66123 Saarbrücken, Germany

**Keywords:** intrinsic healing, magnetic nanoparticles, polymer nanocomposites, water absorption, tensile testing, mechanical characteristics, agglomeration, small-angle X-ray scattering

## Abstract

Self-healing nanocomposites can be generated by organic functionalization of inorganic nanoparticles and complementary functionalization of the polymer matrix, allowing reversible interactions between the two components. Here, we report on self-healing nanocomposites based on ionic interactions between anionic copolymers consisting of di(ethylene glycol) methyl ether methacrylate, sodium 4-(methacryloyloxy)butan-1-sulfonate, and cationically functionalized iron oxide nanoparticles. The materials exhibited hygroscopic behavior. At water contents < 6%, the shear modulus was reduced by up to 90%. The nanoparticle concentration was identified as a second factor strongly influencing the mechanical properties of the materials. Backscattered scanning electron microscopy and small-angle X-ray scattering measurements showed the formation of agglomerates in the size range of 100 nm to a few µm in diameter, independent of concentration, resulting in the disordering of the semi-crystalline ionic polymer blocks. These effects resulted in an increase in the shear modulus of the composite from 3.7 MPa to 5.6 MPa, 6.3 Mpa, and 7.5 MPa for 2, 10, and 20 wt% particles, respectively. Temperature-induced self-healing was possible for all composites investigated. However, only 36% of the maximum stress could be recovered in systems with a low nanoparticle content, whereas the original properties were largely restored (>85%) at higher particle contents.

## 1. Introduction

Inorganic–organic nanocomposites have proven to be high-performance materials due to their outstanding properties, which result from the synergistic combination of properties of the individual components. A general route for their formation is the incorporation of inorganic fillers in a polymer matrix, which significantly improves the mechanical and thermal stability of the matrix material [1,2,3,4]. An interesting aspect regarding sustainability goals in materials development is to increase the lifespan of such materials. Organic functional groups allow the implementation of an even longer lifespan if self-healing properties are induced [5,6,7,8]. While in inorganic materials, healing mechanisms are often associated with high energy inputs [9,10], the concept of inorganic–organic nanocomposites allows the introduction of self-healing into a solid material via organic groups that permit reversible bond formation, e.g., Diels–Alder chemistry (DA), or supramolecular interactions [11,12].

DA chemistry previously proved to have high healing efficiency and cyclability in polymer science [13,14]. We showed in previous work that this type of chemistry can be transferred to the surface of silica or iron oxide nanoparticles [15,16,17]. The self-healing properties of nanocomposites through reversible bond formation in DA chemistry have been demonstrated for a variety of polymer matrices, including epoxy, polyester, polyurethane, methacrylate, and siloxane-based thermoplastics and elastomers [18,19,20,21,22,23,24]. A drawback of DA chemistry on the surface of nanoparticles is the steric demand of the reversible switching process of the DA reaction, which can partially be overcome by tailoring the attachment of the organic-groups to the surface [25]. Higher healing efficiencies and shorter healing times in nanocomposites can be achieved when two independent healing mechanisms are addressed in the same nanocomposite, e.g., hydrogen and coordination bonds, hydrogen bonds and reversible imine/boronate ester bonds, or DA chemistry and reversible hydrogen bonding [26,27,28,29,30].

Besides covalent bond formation and hydrogen bonding, other interaction mechanisms have been addressed in self-healing materials, such as ionic bonding. One of the most studied self-healing materials based on ionic interactions is poly(ethylene-*co*-methacrylic acid), which shows excellent autonomous healing in response to ballistic impact based on the formation of ionic clusters [31,32]. In 2014, Wang et al. first utilized the ionic interactions between poly(ethyleneimine) and poly(acrylic acid) to obtain a self-healing material [33]. Later, poly(allylamine hydrochloride) or poly(triethyl(4-vinylbenzyl)phosphonium chloride) blends with poly(acrylic acid) as well as zwitterionic cross-linked poly(methacrylic acid) and amphoteric polymers were also successfully applied in the field of self-healing [34,35,36,37,38]. All systems showed excellent healing, although the addition of water or salt water is usually required for efficient healing. Despite these promising results, the application of ionic self-healing in nanocomposites has hardly been tested so far. Initial work in this area primarily involved the cationic or anionic modification of silica particles and their subsequent embedding in poly(acrylic acid), imidazolium, or quaternary ammonium functionalized matrices [39,40,41].

For the modification of silica particles, usually alkoxysilane anchor groups are used to attach organic functions important for the healing mechanisms, but phosphonates are better suited for transition metals. They are characterized by a high level of stability and flexibility, have an efficient attachment to the particle surface, and also often have straightforward synthetic accessibility [42,43,44,45,46]. Different functional groups that are of interest for self-healing applications can be combined with these phosphonate anchor groups, such as ligands for complexation, aldehydes or amines for dynamic imine bonds, boronic acid groups for reversible borate ester bonds, cyclodextrins for host-guest interactions, urea groups for hydrogen bonding, DA groups for dynamic carbon-carbon bonds, or stable surface charges for ionic interactions [25,30,42,47,48,49,50,51].

While in many studies the properties of the inorganic nanoparticles were not specifically addressed because the enhancement of the thermal and mechanic properties of the nanocomposites was targeted, this has changed in recent years. Inorganic nanoparticles can induce additional extraordinary properties into the materials. For example, iron oxide (Fe_2_O_3_, Fe_3_O_4_) can induce magnetism in the final material in addition to increasing its thermal stability and improving its tensile strength [52]. Bulk ferromagnetic oxides change to the superparamagnetic state when they fall below a critical size threshold [53]. This loss of permanent magnetization makes these systems particularly attractive, as no agglomeration is expected due to magnetic interactions between the particles. Superparamagnetic iron oxide nanoparticles induce local heat by relaxation when exposed to an alternating magnetic field (AMF) [54]. Due to the induction heating of the iron oxide particles and their straightforward synthesis and low toxicity, they have already found wide applications in the biomedical field [55,56,57,58], whereas their use in self-healing materials is almost unexplored. The first work in this field was carried out by Corten and Urban, in which an alternating electromagnetic field was used for healing a methyl methacrylate matrix via chain interdiffusion [59]. In addition, it could be shown that supramolecular healing mechanisms can be accelerated by inductive heating. Hohlbein et al. described that heating iron oxide nanoparticles in an alternating magnetic field can improve healing based on the formation of zinc acrylate complexes [60]. A comparable improvement was also observed for a gelatin/agarose hydrogel that heals based on hydrogen bonds [61].

In a previous work, we produced self-healing nanocomposites based on ionic interactions, which can be healed in an alternating electromagnetic field [62]. For this purpose, superparamagnetic iron oxide nanoparticles were functionalized with *N*,*N*,*N*-trimethyl-6-phosphonhexan-1-ammonium bromide, introducing positive charges on the particle surface. These were combined with an anionic copolymer of di(ethylene glycol) methyl ether methacrylate (DEGMA) and sodium 4-(methacryloyloxy)butane-1-sulfonate (SMBS) as the organic matrix. By combining the two components, magnetic nanocomposites were obtained, which can be classically cured in an oven and an alternating magnetic field. Although the nanocomposites exhibited efficient self-healing capacity, the initial tests indicated particle agglomeration and an affinity of the polymer system for water.

The aim of this work was the systematic investigation of particle agglomeration and water adsorption within such nanocomposites. Instead of using a controlled radical polymerization, like in previous work, we applied a free-radical polymerization, which allowed us to generate larger amounts of the nanocomposites. A statistical distribution of the negative charges within the polymer should lead to a good dispersion of the particles in the nanocomposite. Small-angle X-ray scattering (SAXS) and backscattered scanning electron microscopy (BSE SEM) were used to determine the particle distribution within the polymer matrix. Due to the affinity of the observed system for water, a drying and storage procedure was developed to make the results comparable for each sample. The water adsorption and its influence on the materials’ properties were investigated with a combination of TGA and torsion/tensile tests. Self-healing experiments in an oven were performed to investigate the influence of the parameters on the healing efficiency.

## 2. Materials and Methods

### 2.1. Materials

All components used for polymerization were degassed before the start of the reaction. The commercially obtained DEGMA was purified and freed from the inhibitor by distillation and using an aluminum oxide column (active basic). The 2,2′-azobis(2-methylpropionitrile) (AIBN) was recrystallized prior to use.

Tris(2,4-dioxopentan-3-ido-κ^2^O,O′)iron (Fe(acac)_3_) (97%), benzyl ether (98%), 1,2-dodecandiol (90%), oleic acid (≥99%), oleyl amine (70%), di(ethylene glycol) methyl ether methacrylate (95%), 1,6-dibromhexane (96%), triethyl phosphite (98%), trimethylamine solution in ethanol (31–35%), and bromtrimethylsilane (97%) were purchased from Sigma-Aldrich (St. Louis, MI, USA). Methacrylic acid (≥99%) was purchased from Merck (Rahway, NJ, USA). Aluminum oxide 90 active neutral (70–230 mesh ASTM) was purchased from Merck Millipore (Burlington, MA, USA). 2,2′-Bipyridine (>99%) was purchased from TCI Chemicals Germany (Eschborn, Germany). Ethyl 2-bromoisobutyrate = EBiB (98%) and hydrazine monohydrate (98%) were purchased from Alfa Aesar (Ward Hill, MA, USA). Sodium bicarbonate (>99%) was purchased from Fluka (Buchs, Switzerland). Hydrochloric acid (37%) was purchased from Bernd Kraft GmbH. Dry dichloromethane (99.8%) and methanol (99.8%) were purchased from Acros Organics (Antwerp, Belgium). Pre-wetted cellulose regenerated tubing membranes with a 3500 D MWCO were purchased from Fisher Scientific (Waltham, MA, USA).

### 2.2. Characterization

Fourier transform infrared (FTIR) spectra were recorded from 4500–400 cm^−1^ with a Bruker Vertex 70 spectrometer (Bruker Optics, Ettlingen, Germany) under ambient air (16 scans at a resolution of 4 cm^−1^) in attenuated total reflectance (ATR) mode.

Differential scanning calorimetry (DSC) measurements were performed using a Netzsch DSC 204 F1 Phoenix (Netzsch–Gerätebau GmbH, Selb, Germany). Samples were prepared in aluminum crucibles with pierced lids and heated under nitrogen at a rate of 5 K/min, 10 K/min, or 20 K/min.

Solution NMR spectra were recorded with a Bruker Avance III HD 300/400 spectrometer (Bruker, Billerica, MA, USA) at 25 °C (^1^H at 300/400 MHz, ^13^C at 75/101 MHz, ^31^P at 121/162 MHz) using CDCl_3_ or D_2_O as solvents and the residual protons of the solvent and carbon as references.

Elemental analysis was performed with a Vario Micro Cube from Elementar (Elementar Analysensysteme GmbH, Langenselbold, Germany).

A Netzsch TG 209 F1 Iris (Netzsch–Gerätebau GmbH, Selb, Germany) was used for thermogravimetric analysis (TGA). Measurements were conducted in aluminum oxide crucibles, heating from room temperature to 880 °C under a nitrogen atmosphere, followed by heating to 1000 °C under a mixture of nitrogen and oxygen (4:1) at a rate of 10 K/min. For determination of the water content in the samples, aluminum crucibles were used, and the lids were pierced immediately before measurement to avoid further water adsorption from the ambient air.

Backscattering scanning electron micrographs were measured with a JEOL JSM-7000 F microscope (JEOL GmbH, Freising, Germany) operating at 20 kV as used for electron acceleration and a working distance of 10 mm. The SEM samples were prepared by placing thin films of the composites on a specimen stub covered with a carbon adhesive foil, followed by the deposition of a gold layer (JEOL JFC-1300 auto fine coater, 30 mA, 40 s) to avoid charging effects. Microscope images were recorded with an Axioskop 50 transmitted light/fluorescence microscope with an Axio-Cam MRc (Carl Zeiss Microscopy GmbH, Oberkochen, Germany).

Powder X-ray diffraction (PXRD) patterns were recorded with a Bruker D8 Advance diffractometer (Bruker AXS, Karlsruhe, Germany) in a Bragg-Brentano *θ-θ*-geometry (goniometer radius 280 mm). A 2*θ* range from 7 to 120° (step size 0.013°) was recorded in a 1 h scan time. Cu-K_α_ radiation (λ = 154.0596 pm, 40 kV, 40 mA) was used with a 12 μm Ni foil to reduce K_β_ radiation. A variable divergence slit was mounted at the primary beam side (irradiated sample area: 10 × 7 mm). A LYNXEYE 1D detector was used on the secondary beam side. Background caused by white radiation and sample fluorescence was reduced by limiting the energy range of the detection. Interpretation of the XRD data was performed via Rietveld analysis using TOPAS 5. The crystallographic structure and micro-structure were refined, while instrumental line broadening was included in a fundamental parameters approach. The mean crystallite size <L> was calculated at the mean volume-weighted column height derived from the integral breadth. The background of standard measurements was fitted by a Chebyshev polynomial function of the 15th degree.

Dynamic light scattering (DLS) studies were performed on an ALV/CGS-3 (ALV GmbH, Langen, Germany) compact goniometer system with an ALV/LSE-5003 correlator at a 90° goniometer angle, using noninvasive backscattering (λ = 632.8 nm). Samples from the reaction solution were diluted, homogenized in an ultrasound bath, and equilibrated for about five minutes before measurement.

Shear, storage, and loss moduli of the samples were determined with a MCR 301 modular compact rheometer (Anton Paar Germany GmbH, Ostfildern–Scharnhausen, Germany). Oscillation measurements with an amplitude of 0.05% and a frequency of 1 Hz were carried out. The specimens were held under tension with a force of 1 N. The samples were heated to 52.5 °C and then cooled to room temperature. The parameters given above were determined at 50 °C.

Small-angle X-ray scattering experiments were performed on a Xenocs Xeuss 2.0 system (Xenocs SA, Grenoble, France). A collimated X-ray beam from a Copper K_α_ source (wavelength λ = 1.54 Å) was used to irradiate the sample, focused on a split size of 0.25 mm^2^. 2D scattering intensity patterns were recorded using a Pilatus 300 K detector from Dectris (Baden, Switzerland) with pixel sizes of 172 × 172 μm^2^ at a sample-to-detector distance of ~1050 mm, calibrated using a silver behenate standard. All samples scattered purely isotropic. Therefore, the 2D scattering patterns were azimuthally averaged to obtain the *q*-dependent scattered intensity, *I*(*q*), where *q* is the absolute value of the momentum transfer, given by *q* = 4π × sin(*θ*/2)/λ, and *θ* is the scattering angle. All samples were measured for 1800 s.

Quasi-static uniaxial tensile tests were performed under displacement control at a constant temperature of 20 °C with a strain rate of 5 × 10^−3^ s^−1^ using a custom-built uniaxial testing device equipped with a heat chamber. In order to obtain the desired uniform temperature, the samples were pre-heated in the unstressed state for 15 min before they were studied under tensile loading. Tests were performed until the fracture of the specimens. Dog bone samples were prepared in a hot press at 80 °C using Teflon molds. Samples with an initial length, width, and thickness of 27 mm, 3 mm, and 2 mm, respectively, were used. A waisted sample geometry (R60) was used whereby a smallest cross-section area in the center of the specimen with the highest stresses is obtained, which generates a replicable predetermined breaking point [63].

### 2.3. Syntheses

The synthesis of the cationic particles and the SMBS were performed as previously described [62]. A short summary of the synthesis protocol and characterization of the products is shown in the Appendix A.

Optimization of polymerization conditions:

Polymerization with AIBN: Initially, kinetic studies were performed to evaluate the ideal reaction time. SMBS and DEGMA (n_ges_ = 32.7 mmol) were placed in a 100 mL two-necked flask in the desired ratio and dissolved in 43 mL water and 9 mL methanol. Pyridine (1 mL) was added as a tracer. The reaction mixture was flushed with argon for 10 min. Then, 1 mol% AIBN was added. The reaction mixture was flushed with argon for a further 10 min before heating to 70 °C. After 30 min, 1 h, 2 h, 3 h, 4 h, and 5 h of heating, a sample was taken each time and analyzed with ^1^H NMR spectroscopy. Studies on the initiator concentration were performed under the same conditions.

Polymerization with DBPO: SMBS and DEGMA were placed in a 100 mL two-necked flask in the desired ratio (n_ges_ = 32.7 mmol) and dissolved in 43 mL water and 9 mL methanol. Pyridine (1 mL) was added as a tracer. The reaction mixture was flushed with argon for 10 min. Then, 1 mol% DBPO was added. The reaction mixture was flushed with argon for a further 10 min before heating to 85 °C. After 30 min, 1 h, 2 h, 3 h, 4 h, and 5 h of heating, a sample was taken each time and analyzed with ^1^H NMR spectroscopy. Temperature studies were performed under the same conditions.

Optimized Polymerization Procedure: SMBS and DEGMA (n_ges_ = 65.4 mmol) were placed in a 250 mL two-necked flask in the desired ratio and dissolved in 86 mL water and 18 mL methanol. The reaction mixture was flushed with argon for 20 min. Then, 1 mol% DBPO was added. The reaction mixture was flushed with argon for a further 20 min before heating to 60 °C for 5 h. All polymer samples were dried under vacuum at 40 °C for two days, then 5 h at 80 °C, and then stored in a desiccator over phosphorus pentoxide for at least two weeks.

Studies on water uptake: The storage under increased humidity took place in a desiccator over water. To further increase the humidity, the desiccator was initially vacuumed to 600 mbar. The water content of the samples was determined after 1 h, 2 h, 4 h, 8 h, 16 h, 32 h, and 64 h with TGA.

Studies on water removal: The samples were placed in an oven at 80 °C for 2 h, 5 h, 8 h, and 24 h. After that, the samples were immediately placed in closed, cold-welded crucibles, which were pierced right before TGA to avoid reuptake of water.

Composite synthesis: First, 0.5 g of the polymer was dissolved in 14 mL of a mixture of THF and H_2_O (v:v, 1:1) under ultrasonication. The appropriate amount of functionalized nanoparticles (2 wt.%, 5 wt.%, 10 wt.%, and 20 wt.%) was added. The solvent was slowly removed under stirring. The composite samples were dried under vacuum at 40 °C for two days, then 2 h at 80 °C, and then stored in a desiccator over phosphorus pentoxide for at least two weeks. In a second batch, the 80 °C drying step was skipped.

Sample preparation: Polymer and composite films were prepared by compression molding at 80 °C for 24 h in a Teflon form, which was held using a vice, if not noted otherwise.

Self-healing tests: Self-healing experiments were performed by cutting the samples through a half thickness. Afterwards, the edges were gently pressed together at room temperature. Then, the samples were heated to 80 °C for 24 h. Afterwards, the samples were put back into the Teflon molds and heated to 80 °C for another 24 h.

## 3. Results and Discussion

### 3.1. General Synthesis of the Self-Healing Nanocomposites

The structure of the synthesized composites is shown in Figure 1.

The cationic groups on the particle surface and the anionic groups in the polymer should have as stable a charge as possible, preferably independently of environmental conditions such as pH. In addition, the polymers should exhibit low glass transition temperatures (*T*_g_), which is advantageous for self-healing. Cationic iron oxide nanoparticles with a size of 6.0 ± 0.9 nm were synthesized and functionalized with the organophosphorus molecule *N*,*N*,*N*-trimethyl-6-phosphonhexan-1-ammonium bromide as described in earlier works [51,62]. Their successful synthesis and functionalization with the cationic phosphonic acid are shown in the Appendix A using FTIR spectroscopy, XRD, TEM, DLS, and TGA (Appendix A). Methacrylate-based monomers were chosen because of their high reactivity and flexibility due to their easy synthetic accessibility. SMBS was used to introduce the charges to the polymer. DEGMA was chosen as a polar low *T*_g_ comonomer. The anionic polyelectrolytes were synthesized via free radical polymerization. The functionalized nanoparticles and the polymers were mixed and stirred in a water–THF mixture. After evaporation of the solvent, the nanocomposites were obtained.

### 3.2. Polymer Synthesis

Three main parameters were considered for polymer synthesis: first, the polymers obtained must be soluble in order to characterize the samples and ensure processability. Second, the polymer composition must be adjustable since this largely determines the material properties and thus its range of applications; and third, the synthesis must be scalable to meet the material requirements of the mechanical measurements. Therefore, in contrast to earlier studies, we have chosen free radical polymerization (FRP) instead of atom transfer radical polymerization (ATRP) for the production of the polymer matrix. Still, FRP also required some optimization to obtain polymers that could be at least partially solubilized to allow characterization and synthesis of the nanocomposites. The summary of the experimental data of the optimization of the polymerization process is shown in the Appendix A. SMBS:DEGMA monomer ratios of 1:3, 1:5, 1:8, and 1:10 were studied to investigate the influence of the polymer composition on its mechanical and thermal properties as well as its solubility and water absorption behavior. Unreacted monomers were removed via dialysis. The IR spectra of the obtained polymers are shown in Figure 1a.

The IR spectra were normalized to the asymmetric C–O stretching vibration (1107 cm^−1^). All spectra reveal the C–H (2820–2927 cm^−1^) and C=O (1724 cm^−1^) stretching vibrations of the methacrylate backbone as well as the broad signal of the O–H stretching vibration (3417 cm^−1^), indicating the presence of water. The ^1^H NMR spectrum is shown exemplarily in Figure 1b for the polymer system with an initial monomer ratio of 1:3 (SMBS:DEGMA). The spectra of the other polymers can be found in the Appendix A. The NMR spectra show the expected signals for the copolymer. The SMBS:DEGMA ratio in the polymer was calculated from the integration ratios of the CH_2_ groups adjacent to the methacrylate (4.20 ppm) and the sulfonate group (2.91 ppm). In contrast to the polymers previously produced by ATRP [62], the composition determined by NMR does not match the monomer ratio used (Table 1). Therefore, additional elemental analysis was carried out on the polymers. The elemental compositions are shown in the Appendix A. From the determined mass fraction of sulfur, the mass fraction of SMBS was calculated. The remaining mass was assumed to be DEGMA, allowing for the determination of the SMBS:DEGMA ratio in the polymer (Table 1). The ratios calculated from the CHN data show that the proportion of ionic species is significantly lower than the initial monomer ratio and the ionic content determined by NMR. This is presumably due to the better solubility of the polymers with an increasing content of ionic species in the NMR solvent used (D_2_O). Since the CHN data better reflect the overall polymer composition, the polymers are referred to below as ^22.4^Pol, ^19.5^Pol, ^13.7^Pol, and ^9.4^Pol, in reference to the DEGMA:SMBS ratios determined from the elemental analysis. Further, the CHN data were used to calculate the molar proportion of SMBS in the polymer in percents.

In summary, the requirements for the solubility of the polymers and the scalability of the polymerization could be met to a large extent. In contrast, although the polymer composition could be varied, the SMBS content was below the originally expected range. Nevertheless, the influence of the polymer composition on the polymer properties was subsequently investigated.

### 3.3. Characterization of the Mechanical and Thermal Properties of the Polymers

DEGMA and polyelectrolyte-based polymer systems can adsorb water from ambient air [64,65,66]. The water content can influence properties such as ion and electric conductivity and, for our study most importantly, the mechanical strength of the polymer-based materials [66,67,68,69]. In order to investigate the influence of the water content on the mechanical properties, we applied a combination of TGA and rheology studies. The polymer samples were stored over water in a desiccator at reduced pressure (600 mbar) for different periods of time. TGA measurements were performed after 2 h, 5 h, 8 h, and 24 h. Figure 2a shows the TGA curves obtained as an example for the ^9.4^Pol system. The complete data can be found in the Appendix A. Regardless of the polymer composition, all TGA curves showed a strong decrease in mass between 100 °C and 150 °C with increasing storage time under humidity. At this temperature, the physically adsorbed water is desorbed. To quantify the water content, the mass loss was determined in the range between 25 °C and 200 °C (Figure 2b).

The water content of the samples increased with increasing storage time. After an initial rapid water absorption, the absorption slowed down, reaching almost a saturation, which is commonly described in the literature [64,70,71]. However, in contrast to earlier reports, the saturation concentration did not seem to correlate directly with the number of sulfonate groups [72]. After just two hours, the samples were noticeably softer and swollen.

To quantify this influence, rheology studies were carried out. The dynamic modulus as well as the shear modulus of the samples was determined in oscillation experiments before and after water absorption. The samples were heated to 52.5 °C and then cooled to room temperature (Appendix A). Figure 2c shows the shear modulus at 50 °C as a function of the water content. For all samples, an almost linear decrease of the shear modulus was observed with an increasing water content. The steepness of the decrease in shear modulus increased with the ionic content in the polymer. While a decrease of 42% was observed for the ^9.4^Pol system with a water absorption of 5.9%, the modulus of the ^22.4^Pol system decreased by only 9.5% with a comparable water absorption (5.5%). Overall, a maximum decrease between 61.9% and 90.6% was observed with a maximum storage time of 24 h for the different polymer samples.

The description of the storage and loss moduli as a function of the storage time at elevated humidity is shown in the Appendix A. In general, the storage modulus was larger than the loss modulus, independent of the polymer composition and water content. The stored deformation energy was, therefore, larger than the deformation energy dissipated by internal friction, indicating physicochemical interactions between the polymer strands. These results are typical for viscoelastic solids [73,74]. For both storage and loss modulus, a steep, almost linear decrease was observed with an increasing water content. The storage modulus decreased due to the decreased stiffness. Furthermore, the reduced friction between the polymer chains due to the water led to a decrease in the dissipated heat and, thus, the loss modulus. The results showed that with an increasing ionic content in the polymer, the dynamic modulus decreased faster. For the ^9.4^Pol system, a 56% decrease in the storage modulus and a 65% decrease in the loss modulus were observed for a water uptake of about 5.9%. The ^22.4^Pol system showed a decrease in the storage and loss modulus of 7% and 20%, respectively, at a comparable water uptake (5.5%). The dissipation factor, which is calculated from the ratio of the loss modulus and storage modulus, was used to classify the change in material properties. A plot of the dissipation factor against the amount of water contained is shown in Figure 2d. In general, the polymers with a higher ionic content showed an initial higher dissipation factor. All samples showed a significant decrease in the dissipation factor with increasing water content. A decrease in dissipation factor corresponded to a shift toward an ideal elastic body. The adsorbed water acts as a plasticizer, as it increases the chain mobility due to its ability to reduce internal interactions, such as hydrogen bonding or electrostatic interactions between polymer chains while increasing molecular spaces [75].

Combining the TGA data and rheology measurements showed that the mechanical properties of the polymer systems were highly dependent on the amount of water absorbed from the ambient air. Even minor differences in the water content significantly changed the mechanical properties and flow behavior of the polymers drastically. The greatest drop in dynamic modulus was always observed in the first few hours of storage at high humidity. After 4.5 h, the dynamic modulus decreased only slightly, which is due to the then-decreasing water uptake, rather than a reduced influence of the absorbed water, as an almost linear relationship between the water content and dynamic modulus can still be observed thereafter. Therefore, an optimized drying and storage procedure for the polymers is required for reproducible characterization of the mechanical properties of these materials.

For this purpose, an analogous series of measurements was carried out, investigating the influence of the storage time at 80 °C on the water content of the polymer samples to evaluate the time when the comprised water is completely removed. At storage times longer than 8 h, the polymers turned from colorless to pale yellow, indicating a partial decomposition. After drying, the samples were analyzed using TGA (Appendix A). The measurements showed only small overall differences in water content, visible in the low-temperature range (<200 °C), in which the adsorbed water is removed. To avoid decomposition and since only small amounts of residual water were observed after two hours at 80 °C, the samples for the following studies were dried for two hours at 80 °C. Thereafter, they were stored in a desiccator at reduced pressure (15 mbar) over phosphorus pentoxide for at least one week prior to use.

Figure 3 shows the shear moduli determined at 50 °C as a function of the polymer composition after the optimized drying procedure. The molar proportion of SMBS in the polymer was calculated from the CHN data. The shear modulus increased linearly with the proportion of ionic groups in the polymer, which is expected as an ionic homopolymer shows a much higher rigidity than a DEGMA-based homopolymer. The results show that the shear modulus can be tailored by the composition of the polymer for a variety of applications.

In the next step, the thermal properties of the dry polymers were investigated, as they are known to be vastly influenced by the polymer composition as well [62]. This was investigated by TGA and DSC measurements of the dry polymers. The experimental data are shown in the Appendix A. The TGA curves indicated no water loss in the range below 200 °C, which confirmed the successful drying. Irrespective of the composition of the polymers, decomposition started above 250 °C. The degradation was primarily defined by the degradation of the DEGMA, while the degradation of the SMBS can only be seen as a shoulder at about 375 °C. The copolymers decomposed almost completely at 900 °C under a nitrogen atmosphere (char yield < 6.5%). After switching to synthetic air, the residual mass dropped to less than 3%. The *T*_95_ values (temperature at 5% mass loss in TGA) were determined to evaluate the thermal stability (Figure 4). Upon increasing the SMBS content from 4.3% to 4.9%, 6.8%, and 9.6%, the *T*_95_ values decreased from 249 °C to 235 °C, 207 °C and 211 °C, respectively. This is surprising, as it was expected that an increase in the more thermally stable SMBS would improve the overall thermal stability. Since this effect was not observed for SMBS-DEGMA copolymers synthesized via ARGET ATRP in a previous report [62], the cause seems to be related to the different polymer structures obtained by the two polymerization techniques. This structure will be discussed in more detail below. However, the exact cause of the effect has not yet been elucidated. As a result, the tailoring of the thermal stability by the polymer composition is rather restricted.

From the DSC studies, the glass transition temperatures of the polymers were determined. This parameter is critical because high chain mobility, which correlates with low *T*_g_s, is essential for high healing efficiencies [24]. DSC measurements showed that the *T*_g_s did not vary much when varying the polymer composition (Figure 4). Values between −35 °C and −29 °C were obtained. This is in line with the observations from previous studies that a certain threshold value of SMBS has to be exceeded in order to bring about significant changes in the glass transition temperature [62]. Nevertheless, a slight increase in *T*_g_ with increasing ionic species was observed. However, the *T*_g_s were about 10 °C lower than for comparable systems synthesized via ATRP, described in earlier studies. The influence of the polymer composition on the *T*_g_ was much lower, so no adjustment over a wide temperature range is possible. No second glass transition was observed in the region of the *T*_g_ of poly(SMBS) (125 °C). Due to the low concentration of SMBS in the polymer and the already low intensity of the glass transitions in the DSC, it cannot be excluded that block-like structures or even separately present ionic and non-ionic polymer strands are present. A more detailed examination of the nature of the polymer was carried out as part of the SAXS analysis.

Like the thermal properties, the tensile properties depend strongly on the polymer composition [76,77]. Uniaxial tensile tests were performed to investigate this influence. The E-modulus, maximum tensile stress (σ_max_), and maximum strain (ε_max_) to failure were determined as mechanical material parameters. The 0.2% yield point (R_p0.2_) was also determined here. First, the reproducibility of the specimens was examined. For this purpose, three specimens were prepared from the ^9.4^Pol system and analyzed. The stress-strain diagrams obtained are shown in Appendix A. In contrast to the rheology study, tensile testing required larger specimens, which are more prone to inhomogeneities. These inhomogeneities in the material vastly influence the results, potentially leading to crack formation and hampering the reproducibility of the measurements. Therefore, the data should be regarded with caution. The stress-strain curves of the synthesized polymers are shown in the Appendix A. The parameters E-modulus, σ_max_, ε_max,_ and R_p0.2_ were determined and are shown in Figure 5 in regard to their dependency on the polymer composition.

E-modulus, σ_max_ and R_p0.2_ showed an almost linear increase with increasing content of the ionic species as expected when increasing the mechanically durable component in the polymer. For the maximum strain, on the other hand, no trend is visible, which is an effect of some pores in the specimens. These results are in good agreement with the observed increase in shear modulus from the rheology tests, confirming that the mechanical properties, unlike the thermal properties, can be well-adjusted via the polymer composition. As the polymers with a lower ionic content are quite soft and the areas of application are therefore rather limited, the stiffest one, the ^9.4^Pol system, was used for further composite synthesis.

### 3.4. Composite Synthesis and Characterization

To produce the composite systems, the ^9.4^Pol was dissolved in a mixture of water and THF. Then, the nanoparticles were added. A total of three different nanoparticle weight fractions were prepared: 2 wt% (NC2), 10 wt% (NC10), and 20 wt% (NC20). The samples were dried under vacuum at 40 °C for two days, then for two hours at 80 °C and stored in a desiccator over phosphorus pentoxide for at least two weeks. In a second batch of samples, the 80 °C drying step was skipped to examine the influence of drying on the agglomeration process of the nanoparticles in the polymer. The composite samples were first examined using IR spectroscopy (Figure 6).

The IR spectra show the expected signals of the polymer. In particular, the signals of the asymmetric C=O and C–O stretching vibrations at 1723 cm^−1^ and 1105 cm^−1^, respectively, as well as the C–H stretching vibrations at 2819–2932 cm^−1^, are clearly visible. In addition, the signal of the Fe–O–Fe vibration of the iron oxide particles appears at 558 cm^−1^.

The composites were also investigated for their thermal properties, as it is known that the integration of an inorganic component often significantly improves the thermal stability and influences the glass transition temperatures [62,78]. The TGA curves of the nanocomposites are shown in the Appendix A. Similar to the pristine polymers, the decomposition of the composites was determined by the DEGMA decomposition and the associated mass loss starting above 250 °C. As no decomposition or sublimation of the particles is expected, the char yields after the nitrogen segment are larger the more particles are included. A residual mass of 17.6% was observed for the NC20 system, while the polymer sample showed only a residual mass of 3.9% at 900 °C. In the subsequent air segment, a further decrease in mass was observed in the composites analogous to the polymers. However, this loss was less pronounced with an increasing particle content. This can be attributed to two aspects. Firstly, a smaller decrease in mass is expected, as a smaller proportion of the sample is polymer, which is further decomposed in this step and accounts for the decrease in mass. Secondly, the particles are oxidized to Fe_2_O_3_, which further counteracts the decrease in mass. As a result, for NC20, even a slight increase in mass was observed upon contact with oxygen. Regarding the thermal stability, again the *T*_95_ values were considered (Figure 7).

Upon integrating 2 wt% particles in the polymer matrix, the *T*_95_ values steeply increased from 212 °C to 225 °C. After that, the thermal stability seemed to reach a plateau at roughly 240 °C. DSC measurements revealed an increase in the *T*_g_s from -33.7 °C to −28.0 °C upon integrating 2 wt% particles (Figure 7). This is consistent with literature reports and is due to the reduced chain mobility resulting from the relatively strong ionic interactions between the particles and the polymer matrix [79,80,81]. A maximum *T*_g_ of −25.4 °C was reached with 20 wt% particle content. No further signals were observed in DSC. The particle content thus represents a set screw for adjusting the thermal properties of nanocomposites. A more detailed discussion of the temperature stability of the polymer matrix can be found in the Appendix A.

In inorganic–organic nanocomposites, the particles ideally should be homogeneously distributed, as agglomerates often act as stress concentration points [82,83,84]. Also, with regard to the inductive healing in an alternating magnetic field, a homogeneous distribution is advantageous in order to produce an even heating in the material. However, agglomeration can occur due to incompatibilities of the particles with the matrix or other agglomeration phenomena. We applied SAXS and BSE SEM measurements to obtain insights into the particle distribution. The samples dried at 40 °C and 80 °C were examined comparatively (Figure 8).

The scattering curve obtained for the sample without nanoparticles showed weak forward scattering and a sharp peak at *q*_0_ = 0.31 Å^−1^. Furthermore, a peak at 2·*q*_0_ was visible (Appendix A). This indicates the presence of a lamellar structure with a repeat distance of 2.1 nm (2π/*q*_0_) [85]. Due to the small increase in the *T*_g_s of the polymers with an increasing number of ionic groups and the absence of a glass transition in the range of poly(SMBS), it seems likely that ionic blocks are responsible for the signal, as a similar effect has been observed for poly(ethylene-*co*-methacrylic acid) [32,86]. The ionic groups of this ionomer tend to aggregate. These so-called multiplets have a spherical shape and consist of fewer than 10 ion pairs, which, in turn, overlap to form paracrystalline structures.

The scattering curves obtained for the samples, which include nanoparticles, showed a broad peak at 0.05–0.1 Å^−1^, revealing that the nanoparticles have well-defined and short inter-distances, which implies that they are agglomerated. At small *q*-values, a strong increase in intensity was observed, which shows that the nanoparticles form agglomerates with sizes of at least 100 nm. Additionally, with an increasing nanoparticle content, the sharp peak at ~0.31 Å^−1^ decreased more strongly in intensity than would be expected from the changing sample composition. Therefore, the initially crystalline areas are broken up due to the preferential coordination of the ionic groups of the polymers to the cationic functionalized particles.

To quantify the structural properties of the nanoparticles in the polymer matrix that are dependent on the particle content and drying temperature, the data were modeled using the function:(1)$$I\left(q\right)=I_{P}\left(q\right)+S_{HS}\left(q\right)P_{pS}\left(q\right)+I_{G}\left(q\right)+I_{bkg}$$

Here, *P*_pS_(*q*) is the form factor of polydisperse spheres following a Gaussian size distribution, which gives the radius *R* of the nanoparticles [87]. *P*_pS_(*q*) is multiplied by a disordered hard-sphere structure factor, *S*_HS_(*q*), accounting for the nanoparticle arrangement [88]. It yields the hard-sphere radius, *R*_HS_, i.e., half of the center-to-center distance of the nanoparticles, and the volume fraction that the hard spheres occupy within the agglomerates, η. Furthermore, a generalized Porod law, *I*_P_(*q*), accounts for large-scale structures with fractal dimension *m*. *I*_G_(*q*) is a Gaussian distribution, accounting for the Bragg peak at ~0.31 Å^−1^, resulting from the paracrystalline structure of the polymers, and *I*_bkg_ is a constant background.

*R*_HS_ (Figure 8c) decreased from 4.3 nm at 2 wt% nanoparticle content to 3.4 nm with an increasing particle content up to 20 wt%, while η (Figure 8d) increased from ~25% to ~33% for the same particle contents. From these observations, we can conclude that the agglomerates densify with increasing particle content for both drying temperatures.

Figure 9a–f shows the BSE SEM micrographs of the composite samples synthesized at 40 °C and 80 °C, respectively. The particles can be recognized as lighter areas in the micrographs. In agreement with the results of the SAXS measurements, agglomerates were observed in all cases, regardless of the particle content and drying temperature. Thus, especially in the composites with 2 wt% nanoparticles, larger areas of the sample were observed where no particles were present at all. While smaller spherical agglomerates with a diameter of about 100 nm were present at low particle concentrations, the size of the agglomerates increased further with an increasing particle concentration up to 10 wt% to a diameter of about 1 µm. The samples prepared at 40 °C appeared less compact. In contrast, the particles seemed to have sintered together after treatment at 80 °C. When the particle content was increased further to 20 wt%, the differences became even more pronounced. In the samples prepared at 40 °C, the size of the agglomerates decreased again, while in the 80 °C sample, the agglomerate size increased even further.

The composites were dried and stored following the same procedure as the polymers. In Figure 10a, the influence of the particle content of the composites on their shear modulus is shown. The shear modulus showed an initial steep increase from 3.72 MPa to 5.61 MPa upon integration of 2 wt% of nanoparticles. Thereafter, a linear increase up to 7.48 MPa was observed for the NC20 system, which is in good agreement with the increase in *T*_g_s as determined by DSC. The samples were then stored under elevated humidity to investigate the water affinity of the materials and the influence of the absorbed water on their mechanical properties. TGA curves at various storage periods at increased humidity, shown exemplarily for the NC2 system in Figure 10b, revealed that the composites also show a high affinity toward water. The water content increased to almost 40% for storage times of 24 h. Figure 10c shows a summary of the results of the moisture absorption study. The speed and amount of water absorption does not correlate with the number of particles in the system. The complete TGA data for the composites stored at high humidity are given in the Appendix A.

The water absorption of the composite was higher than that of the polymer. This is initially surprising since it was expected that the ionic groups would be largely responsible for water absorption and the negative groups in the polymer would be neutralized by the positive charges on the particle surface. However, the fact that this is not the case is consistent with the observations made in the case of the polymer systems, where the content of SMBS in the polymer could not be correlated with the amount of adsorbed water. Therefore, the cause of this effect also seems to originate from the structure of the polymers. Although the temperature-dependent SAXS measurements suggest that water is trapped in the crystalline regions, it is possible that this is less than what could coordinate with the ionic group in an open chain. When these crystalline regions become disordered by adding particles, some ionic groups are neutralized by the particles, but some ionic groups remain uncoordinated in the polymer and are now freely available to adsorb water. This counteracting effect causes the initial amount of water adsorbed to increase when only a few particles are added. With the addition of more particles, the free ionic groups are gradually neutralized by the particles, and the water uptake decreases again.

Subsequently, the influence of water absorption on the mechanical properties was investigated. For this, the shear modulus was examined again (Appendix A). Figure 10d shows the correlation of the shear modulus at 50 °C with storage time at increased humidity. The shear modulus of the composites was also significantly affected by the water content. For all systems, the shear modulus decreased with increasing water absorption. The absorbed solvent also serves as a plasticizer in the composites. As with the polymer, small differences in the water content have a very strong influence on the shear modulus and the mobility of the polymer chains. Therefore, it was investigated whether the increased mobility is also observed for the particles and whether the agglomerate structure changes as a result. For this purpose, SAXS measurements were performed (Appendix A). The overall appearance of the scattering curves is similar to that of the dry systems, meaning that the nanoparticles are strongly agglomerated. As shown in Appendix A, the hard-sphere radius decreased with increasing nanoparticle content, whereas the hard-sphere volume fraction increased, both in the same ranges as for the dry systems. Hence, the agglomerates densify with increasing nanoparticle content. But this process seems neither significantly accelerated nor slowed down by the swelling of the composites due to the absorption of water. Nonetheless, the absorption tests showed that careful drying is the key to reasonable comparative studies for the mechanical properties of composite systems.

### 3.5. Mechanical Characterization of the Composites

Tensile tests were also carried out on the dried composites to study their mechanical properties as a function of particle content. The stress-strain curves are shown in Appendix A. The results are summarized in Figure 11.

The E-modulus increases with the particle content. This is in good agreement with literature reports. Further, it is described that there is also an increase in the E-modulus as the size of the nanoparticles decreases [89]. As a result, the increasing agglomeration of the particles, as observed with NC20, should analogously counteract the increase in the E-modulus, which explains its slight decrease. Both the ultimate tensile stress as well as the ultimate strain showed an almost linear decrease with increasing particle content. The material became harder, but tore faster, as expected from the rheology studies. Jordan et al. described in their review that ultimate tensile stress and strain increase upon strong interactions between filler and matrix and decrease upon weak interactions [89]. This indicates a poor interaction between the polymers and particles used in this study, which is rather surprising. Nonetheless, there are also systems described for which a strong interaction is expected but still, a decrease in yield stress and tensile strength is observed [90,91]. Here, no trends are observable for the R_p0.2_.

### 3.6. Self-Healing Experiments

The composites were evaluated for their self-healing properties. For this, the composites were cut to half their thickness and stored in an oven at 80 °C for 24 h to heal. The healing was examined phenomenologically with a microscope. To determine the efficiency of healing, the specimens were tested in tensile tests after healing. The material constants were then compared before and after the healing process. The microscope images of the healing process for NC2 are shown in Figure 12 and for NC10 and NC20 in the Appendix A.

The microscopy images showed almost complete healing for all composite systems. Only with the NC20 composite could a slight mark still be observed after 48 h. Since chain mobility is key for the healing process, the healing efficiency for the NC20 system might be hampered due to the increased rigidity, as indicated by the increased *T*_g_ and shear modulus. In all cases, a slight discoloration remained at the point of incision. Tensile testing was then used to determine whether the healing process would lead to a decrease in the mechanical stability of the composite. The obtained stress-strain curves of the healed samples in comparison with the initial composites are shown in the Appendix A. The stress-strain curves show a similar pattern for the initial and healed samples. The material parameters determined from this are summarized in Figure 13.

For the maximum elongation, the materials properties are completely recovered upon healing for all composite samples. This is not the case for the maximum tensile stress, E-modulus, and R_p0.2_ at low particle concentration (2 wt%). While for E-modulus and R_p0.2_ a recovery of 69% and 60% was observed, respectively, σ_max_ decreased far more to only 36%. At 10 wt% particle content, all properties were mostly retrieved. For σ_max_, 85%, for R_p0.2_, 90%, and for ε_max_, 100% of the initial properties were recovered. The E-modulus was found to be even slightly higher for the healed sample. Surprisingly, this was also the case for all properties for the NC20 system. While stress and strain at break for the healed samples was very close to the initial values and therefore in the range of the reproducibility errors of the samples, values for the E-modulus and Rp_0.2_ were found as high as 180% of the initial value for the healed samples. As the not-healed samples were subjected to the same temperature profile, different water contents can be ruled out as the reason for the increase of these two parameters. The cause of the observed increase could not be determined yet.

Finally, the particle distribution in the polymer after tensile testing was determined to investigate the influence of the size and shape of the agglomerates. For this, BSE SEM measurements were carried out. The micrographs are shown in Figure 14a–c.

It is noticeable that, especially in the NC10 and NC20 systems, an alignment of the agglomerates takes place. It cannot be determined whether this occurs along the axis of the tensile tests, but it seems likely. In addition, SAXS measurements were carried out on the composites after tensile testing. No signs of anisotropic scattering were observed in the 2D scattering images. Therefore, also in this case, they were azimuthally averaged, and are shown in Figure 14d. The fitting parameters do not differ significantly from those of the dry samples, as is shown in Appendix A. This indicates that, even though the macroscopic orientation of the agglomerates has changed after tensile testing, they remain largely unchanged on length scales probed by SAXS.

In general, all systems could be healed on a µm scale. Healing efficiency was higher at higher particle contents. However, there seemed to be an optimum, since at a certain point the stiffness introduced by the particles impeded healing. This is in the range of 10 wt% to 20 wt%. It was also shown that the mechanical stress led to a change in the agglomerate structure. The extent to which this affects the material properties or multiple healing cycles requires further investigation.

## 4. Conclusions

In this paper, we presented the synthesis of an SMBS-DEGMA copolymer via free radical polymerization. The SMBS:DEGMA ratios in the polymer composition were varied from 1:9.4 to 1:22.4. The polymers were characterized by ^1^H NMR, CHN analysis, FTIR, and DSC, revealing that the incorporation of ionic species in the polymer was lower than expected from the initial monomer ratios of 1:3 to 1:10. Due to the obtained small SMBS contents, *T*_g_s varied only in a 5 °C range. SAXS measurements suggested that block-like structures were obtained where the ionic regions formed crystalline regions that were stable even at high temperatures. All systems showed high affinity toward moisture, showing between 20 wt% and 30 wt% water absorption, reducing the polymers’ shear moduli from 4.58 MPa and 0.49 MPa to 0.43 MPa and 0.19 Mpa, respectively, depending on the polymer composition. Rheology investigations and tensile testing in combination with TGA revealed that the already small amounts of comprised water vastly decreased the polymers’ mechanical properties, with the water functioning as a plasticizer. As a result, a drying and storage procedure was developed to allow comparative studies. The mechanical properties and thermal stability could be improved by incorporating *N*,*N*,*N*-trimethyl-6-phosphonhexan-1-ammonium bromide functionalized iron oxide particles of 6.0 ± 0.9 nm diameter into the polymer matrix. Particle contents of 2 wt%, 10 wt%, and 20 wt% were investigated. Despite the ionic interactions between the polymer matrix and the particles, the synthesized composites showed particle agglomeration by SAXS and BSE SEM, independent of the polymer composition and particle content. Upon integration of the particles in the polymer matrix, the initial crystalline polymer areas became disordered. The materials showed the ability to heal scratches at a µm scale in a conventional oven at 80 °C within 48 h. Tensile testing revealed an almost complete recovery (within measurement accuracy) of the mechanical properties after healing, especially for higher particle contents. The incorporation of superparamagnetic iron oxide nanoparticles allowed for heating in an alternating magnetic field, potentially facilitating spatially resolved healing.

## Data Availability

The data presented in this study are available in Appendix A.

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
