# Peer review of "Self-Healing Iron Oxide Polyelectrolyte Nanocomposites: Influence of Particle Agglomeration and Water on Mechanical Properties"

_nanomaterials, 2023, doi:10.3390/nano13232983_

Round 1

Reviewer 1 Report

Comments and Suggestions for Authors

The paper presents "Self-healing iron oxide polyelectrolyte nanocomposites: Influence of particle agglomeration and water on mechanical properties". The manuscript is a very interesting work and the authors show many unique results. According to my opinion, the authors are suggested to address the following comments in order to meet the requirements of the Journal.

1. Provide the full spelling of AIBN. All abbreviations should be provided in their full spelling on their first appearance

2. A special phenomenon is mentioned in the paper that the increase in the more thermally stable SMBS rather decreases the thermal stability of SMBS-DEGMA copolymers, for which the authors should give a more reasonable explanation.

3. The water absorption of composites is higher than that of polymers as mentioned in the article, for which the authors do not give an explanation.

4. Inconsistent formatting of references. For example, some names have spaces added and some are not (e.g. ref.3); some references do not have page numbers added. Revise this section to meet the requirements of the journal.

5. The formatting of the text in lines 585-593 should be set to "aligned at both ends".

Comments on the Quality of English Language

Quality of  language is fair.

Author Response

  1. Provide the full spelling of AIBN. All abbreviations should be provided in their full spelling on their first appearance

Response: The full spelling of AIBN was added where the term first appears.

  1. A special phenomenon is mentioned in the paper that the increase in the more thermally stable SMBS rather decreases the thermal stability of SMBS-DEGMA copolymers, for which the authors should give a more reasonable explanation.

Response: In contrast to the polymer systems reported here, previously published results on SMBS-DEGMA copolymers showed an increase of thermal stability with increasing SMBS content (Ref. 62). The polymers in the previous study did not show a crystalline fraction (unpublished results), so the difference seems to be due to the polymer structure. The exact cause of this effect has not yet been identified and is beyond the scope of this publication. A reference to the effect of polymer structure has been added at the relevant paragraph.

 “Since this effect was not observed for SMBS-DEGMA-copolymers synthesized via ARGET ATRP in a previous report [62], the cause seems to be related to the different polymer structures obtained by the two polymerization techniques. This structure will be discussed in more detail below. However, the exact cause of the effect has not yet been elucidated.”

  1. The water absorption of composites is higher than that of polymers as mentioned in the article, for which the authors do not give an explanation.

Response: A paragraph was added discussing a possible explanation for this effect.

 “Therefore, the cause of this effect also seems to originate from the structure of the polymers. Although the temperature dependent SAXS measurements suggest that water is trapped in the crystalline regions, it is possible that this is less than what could coordinate to the ionic group in an open chain. When these crystalline regions become disordered by adding particles, some ionic groups are neutralized by the particles, but some ionic groups remain uncoordinated in the polymer and are now freely available to adsorb water. This counteracting effect causes the initial amount of water adsorbed to increase when only a few particles are added. With the addition of more particles, the free ionic groups are gradually neutralized by the particles and the water uptake de-creases again.”

  1. Inconsistent formatting of references. For example, some names have spaces added and some are not (e.g. ref.3); some references do not have page numbers added. Revise this section to meet the requirements of the journal.

Response: All citations were revised and adjusted.

  1. The formatting of the text in lines 585-593 should be set to "aligned at both ends".

Response: The formatting was adjusted.

Reviewer 2 Report

Comments and Suggestions for Authors

The MS by B. Oberhausen relates on a well-perfomed investigation of self-healing nanocomposites based on anionic copolymers and cationically functionalized iron oxide nanoparticles. The MS is well-presented and well-written. The conclusions are fully supported by the experimental data gained through a thorough characterizaation of the nanomaterials. 

However, from a personal perspective, I find the manuscript to be somewhat lengthy, and the novelty of the research may not be immediately apparent.

Author Response

The MS by B. Oberhausen relates on a well-perfomed investigation of self-healing nanocomposites based on anionic copolymers and cationically functionalized iron oxide nanoparticles. The MS is well-presented and well-written. The conclusions are fully supported by the experimental data gained through a thorough characterizaation of the nanomaterials. 

However, from a personal perspective, I find the manuscript to be somewhat lengthy, and the novelty of the research may not be immediately apparent.

Response: Introduction and Results and Discussion sections have been shortened. Shortened sections are marked. These include the literature review on self-healing ionic polymers and silica-based nanocomposites and the NMR and CHN characterization of the ionic polymers. Parts of the discussion have been deleted or moved to the Supporting Information. These include some redundant statements, as well as overdetailed parts of the discussion on rheology, tensile testing and BSE SEM measurements.

Reviewer 3 Report

Comments and Suggestions for Authors

The article by Oberhausen et al. reports the self-healing iron oxide polyelectrolyte nanocomposites and evaluates the influence of particle aggregation and water on mechanical properties. The manuscript is well-drafted and can be published after small corrections. 

·      The IUPAC name of the Fe(acac)3 must be provided in line 144.

·      Instead of mentioning that the information on the synthesis procedure of the cationic particles and the SMBS 229 can be found elsewhere [62], the authors are suggested to provide a brief protocol in supporting information.

·      In line 277, the scheme number needs to be included.

Author Response

The article by Oberhausen et al. reports the self-healing iron oxide polyelectrolyte nanocomposites and evaluates the influence of particle aggregation and water on mechanical properties. The manuscript is well-drafted and can be published after small corrections

The IUPAC name of the Fe(acac)3 must be provided in line 144.

Response: The IUPAC name was added.

Instead of mentioning that the information on the synthesis procedure of the cationic particles and the SMBS 229 can be found elsewhere [62], the authors are suggested to provide a brief protocol in supporting information.

Response: The synthetic procedure was added to the Supporting Information. The references in the text were adjusted accordingly.

In line 277, the scheme number needs to be included.

Response: The Scheme number was added.